# ADAM IS NO BETTER THAN NORMALIZED SGD: DISSECTING HOW ADAPTIVE METHODS IMPROVE GANS PERFORMANCE

## ABSTRACT

Adaptive methods are widely used for training generative adversarial networks (GAN). While there has been some work to pinpoint the marginal value of adaptive methods in minimization problems, it remains unclear why it is still the method of choice for GAN training. This paper formally studies how adaptive methods help performance in GANs. First, we dissect Adam—the most popular adaptive method for GAN training—by comparing with SGDA the direction and the norm of its update vector. We empirically show that SGDA with the same vector norm as Adam reaches similar or even better performance than the latter. This empirical study encourages us to consider normalized stochastic gradient descent ascent (nSGDA) as a simpler alternative to Adam. We then propose a synthetic theoretical framework to understand why nSGDA yields better performance than SGDA for GANs. In that situation, we prove that a GAN trained with nSGDA provably recovers all the modes of the true distribution. In contrast, the same networks trained with SGDA (and any learning rate configuration) suffers from mode collapsing. The critical insight in our analysis is that normalizing the gradients forces the discriminator and generator to update at the *same pace*. We empirically show the competitive performance of nSGDA on real-world datasets.

## 1 INTRODUCTION

It is commonly accepted that adaptive algorithms are required to train modern neural network architectures in various deep learning tasks. This includes minimization problems that arise in natural language processing (Vaswani et al., 2017) and fMRI (Zbontar et al., 2018) or min-max problems such as generative adversarial network (GAN) training (Goodfellow et al., 2014). Indeed, it has been empirically observed that Adam (Kingma & Ba, 2014) yields a solution with better generalization than stochastic gradient descent (SGD) in these problems (Choi et al., 2019). Several works have attempted to explain this phenomenon in the minimization case. Common explanations are that adaptive methods train faster (Zhou et al., 2018), escape faster very flat saddle-point like plateaus (Orvieto et al., 2021) or deal better with heavy-tailed stochastic gradients (Zhang et al., 2019b). However, much less is known regarding min-max problems such as GANs. In this paper, we investigate why GANs trained with adaptive methods outperform those trained using stochastic gradient descent ascent with momentum (SGDA).

Some prior works attribute this outperformance to the superior convergence speed of adaptive methods. For instance, Liu et al. (2019) show that a variant of Optimistic Gradient Descent (Daskalakis et al., 2017) converges faster than SGDA for a class of non-convex non-concave min-max problems. However, contrary to the minimization setting, convergence to a stationary point is not guaranteed and not even a requirement to ensure a satisfactory GAN performance. Indeed, Mescheder et al. (2018) empirically shows that popular architectures such as Wasserstein GANs (WGANs) (Arjovsky et al., 2017) do not always converge, and yet produce realistic images. We support this observation through the following experiment. We train a DCGAN (Radford et al., 2015) using Adam –the most popular adaptive method– and set up the generator ($G$) and discriminator ($D$) step-sizes respectively as $\eta_D, \eta_G$. Note that $D$ is usually trained faster than $G$ i.e. $\eta_D \geq \eta_G$. Figure 1(a) displays the GAN convergence measured by the ratio of gradient norms, and the GAN's performance measured in FID score (Heusel et al., 2017). We observe that when $\eta_D/\eta_G$ is close to 1, the algorithm does not converge and yet, the model produces high-quality solutions. On the other hand, when $\eta_D/\eta_G \gg 1$, the model converges to an equilibrium –a similar statement has been proved by Jin et al. (2020) and Fiez & Ratliff (2020) in the case of SGDA. However, the GAN produces low-quality solutions at

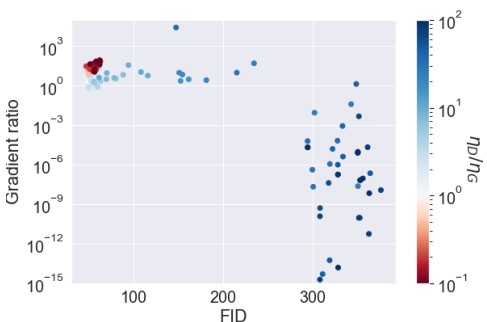

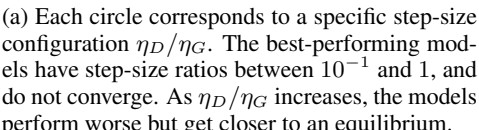

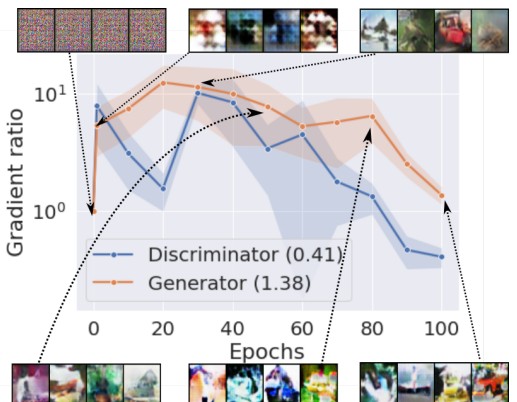

(a) Each circle corresponds to a specific step-size configuration $\eta_D/\eta_G$. The best-performing models have step-size ratios between $10^{-1}$ and 1, and do not converge. As $\eta_D/\eta_G$ increases, the models perform worse but get closer to an equilibrium.

(b) shows that during training, the gradient ratio of a well-performing GAN approximately stays constant to 1. We also display the images produced by the model during training.

Figure 1: Gradient ratio against FID score (a) and number of epochs (b) obtained with DCGAN on CIFAR-10. This ratio is equal to $\|\text{grad}_G^{(t)}\|_2/\|\text{grad}_G^{(0)}\|_2 + \|\text{grad}_D^{(t)}\|_2/\|\text{grad}_D^{(0)}\|_2$, where $\text{grad}_G^{(t)}$ (resp. $\text{grad}_D^{(t)}$) and $\text{grad}_G^{(0)}$ (resp. $\text{grad}_D^{(0)}$) are the current and initial gradients of $G$ (resp. $D$). Note that $\|\cdot\|_2$ refers to the sum of all the parameters norm in a network. For all the plots, the models are trained for 100 epochs using a batch-size 64. For (b), the results are averaged over 5 seeds.

this equilibrium. Thus, simply comparing the convergence speed of adaptive methods and SGDA *cannot* explain the GAN's performance obtained with adaptive methods. This observation motivates the central question in this paper: *What factors explain that Adam produces better quality solutions than SGDA when training GANs?*

To address this question, we dissect Adam following the approach by Agarwal et al. (2020). They frame a generic optimizer's update as $W^{(t+1)} = W^{(t)} - \eta a^{(t)}\mathcal{G}^{(t)}$, where $W^{(t)} \in \mathbb{R}^d$ is the iterate, $\mathcal{G}^{(t)} \in \mathbb{R}^d$ such that $\|\mathcal{G}^{(t)}\|_2 = 1$ is the optimizer's *direction* and $a^{(t)} \geq 0$ is the optimizer's *magnitude*. Therefore, a first step in our paper is to understand whether Adam outperforms SGDA maintly due to its direction or to its magnitude. As detailed in Section 2, we train a GAN using i) AdaLR, an algorithm that updates in the direction of SGDA but with the magnitude of Adam ii) AdaDir which uses the direction of Adam but the magnitude of SGDA. We empirically show that not only, AdaLR significantly outperforms AdaDir, SGDA, and Adam itself. This observation encourages us to conclude that:

*Adam produces higher quality solutions relative to SGDA in GANs mainly due to its adaptive magnitude and not to its adaptive direction.*

In Section 2, we empirically analyze the adaptive magnitude of AdaLR and observe that it stays approximately constant throughout training. This observation eventually encourages the study of AdaLR with a constant step-size. Such algorithm actually corresponds to *normalized* SGDA (nSGDA).Compared to SGDA, nSGDA has the same direction but differs in magnitude since we divide the gradient by its norm. Intuitively, this normalization forces $D$ and $G$ to be updated by vectors with constant magnitudes *no matter how different* the norms of $D$'s and $G$'s gradients are.

Motivated by the aforementioned observations, this paper studies the performance of GANs trained with nSGDA. We believe that this is a first step to formally understand the role of adaptive methods in GANs. Our contributions are divided as follows:

– In Section 3, we experimentally confirm that nSDGA consistently competes with Adam and outperforms SGDA when using different GAN architectures on a wide range of datasets.
– In Section 4, we provide a theoretical explanation on why GANs trained with nSGDA outperform those trained with SGDA. More precisely, we devise a data generation problem where the target distribution $\mathcal{D}$ is made of multiple modes. The model trained with nSGDA provably recovers all the modes in the target distribution while the SGDA's one fails to do it under *any step-size configuration*: We prove that even when SGDA converges to a locally optimal min-max equilibrium, the model still *suffers from mode collapsing* and fails to learn recover the modes separately.

The key insight of our theoretical analysis is that no matter how the step-sizes are prescribed, $D$ and $G$ necessarily update at very different speeds when we use SGDA. Therefore, i) either $D$ updates its weights too fast, thus learns a weighted average of the modes of $\mathcal{D}$. This makes $G$ learn this

weighted average of modes ii) or $D$ does not update its weights fast enough and thus, $G$ aligns its weights with those of $D$. This forces $D$ to converge to a locally optimal min-max equilibrium that classifies any instance as "fake". On the other hand, by normalizing the gradients as done in SGDA, we *force* $D$ and $G$ to *update at the same speed throughout training*. Thus, whenever $D$ learns a mode of the distribution, $G$ learns it right after, which makes both of them learn all the modes of the distribution separately.

Our paper advocates for the use of *balanced updates* in GAN training i.e. the ratio of $D$ vs $G$ updates should remain close to constant. To our knowledge, we are the first to *theoretically* show the importance of these balanced updates. This insight contrasts with the related work that analyzes GANs, and more generally zero-sum differentiable games, in the regime where $D$ is updated much faster than $G$ i.e. $\eta_D/\eta_G \gg 1$ (Fiez & Ratliff, 2020; Jin et al., 2020; Fiez et al., 2020).

### RELATED WORK

**Adaptive methods in games optimization.** Several works designed adaptive algorithms and analyzed their convergence to show their benefits relative to SGDA. For variational inequality problems, Gasnikov et al. (2019); Antonakopoulos et al. (2019); Bach & Levy (2019); Antonakopoulos et al. (2020) propose adaptive algorithms that reach optimal convergence rates under regularity assumptions. For a class of non-convex non-concave min-max problems, Liu et al. (2019); Barazandeh et al. (2021) design algorithms that converge faster than SGDA. On the other hand, Heusel et al. (2017) show that Adam locally converge to a Nash equilibrium in the regime where the step-size of the discriminator is much larger than the one of the generator. Our work differs from these papers as we analyze Adam and do not focus on the convergence properties but rather on the fit of the trained model on the *true* (and not empirical) data distribution. Besides, contrary to some of the aforementioned papers, our work is not in the two-time scale learning-rates regime which do not correspond to what is mostly done in practice.

**Importance of balanced updates in GANs.** The importance of balanced updates for GANs has been noticed in the literature. The most popular GAN architectures (Radford et al., 2015; Arjovsky et al., 2017; Brock et al., 2018) set the step-sizes such that $\eta_D/\eta_G$ is constant. On the other hand, Berthelot et al. (2017) introduced a control variable to ensure that the updates between $G$ and $D$ are at the same speed. In this work, we do not modify the GAN objective loss to enforce balancedness. Instead, we empirically and theoretically investigates how Adam and nSGDA enforce balanced updates.

**Statistical results in GANs.** Early works investigate whether GANs memorize the training data or actually learn the distribution (Arora et al., 2017; 2018; Dumoulin et al., 2016). Zhang et al. (2017); Bai et al. (2018) then show that for specific GANs, the model learn some distributions with non-exponential sample complexity (Liang, 2017; Feizi et al., 2017). Recently, Li & Dou (2020); Allen-Zhu & Li (2021) further characterized the distributions learned by the generator. On the other hand, some works attempted to explain GAN performance through the optimization lens. Lei et al. (2020); Balaji et al. (2021) show that GAN models trained with SGDA converge to a global saddle point when the generator is one-layer neural network and the discriminator is a specific quadratic/linear function. Our contribution significantly differs from these two works as i) we construct a setting where SGDA converges to a locally optimal min-max equilibrium and yet suffer from mode collapse. Conversely, nSGDA does not necessarily converge and yet, recovers the true distribution ii) our setting is more challenging since we need at least a degree-3 discriminator to learn the distribution – see Section 4 for a justification.

**Normalized gradient descent.** Introduced by Nesterov (1984), normalized gradient descent has been widely used in the minimization setting. Indeed, it has been observed that normalizing out the gradient improves the 'slow crawling' problem of gradient descent and avoids the iterates to be stuck in flat regions – such as spurious local minima or saddle points – (Hazan et al., 2015; Levy, 2016; Murray et al., 2019). Normalized gradient descent or its variants outperform the non-normalized counterparts in multi-agent coordination (Cortés, 2006) and deep learning tasks (You et al., 2017; 2019; Cutkosky & Mehta, 2020; Liu et al., 2021). Our work rather considers the min-max setting and shows that nSGDA performs better than SGDA as it forces the discriminator and generator to update at the same rate.

## 2 FROM ADAM TO NSGDA

**Generative adversarial networks.** Given a training set sampled from some target distribution $\mathcal{D}$, a GAN learns to generate new data from this distribution. The architecture is constituted of two

---

**Algorithm 1** Generic second-moment adaptive optimizer

---

**Input**: initial points $\mathcal{W}^{(0)}, \mathcal{V}^{(0)}$, step-size schedules $\{(\eta_G^{(t)}, \eta_D^{(t)})\}$, hyperparameters $\{\beta_1, \beta_2, \varepsilon\}$.

Initialize $\mathbf{M}_{\mathcal{W},1}^{(0)}, \mathbf{M}_{\mathcal{W},2}^{(0)}, \mathbf{M}_{\mathcal{V},1}^{(0)}$ and $\mathbf{M}_{\mathcal{V},2}^{(0)}$ to zero.

**for** $t = 0 \ldots T-1$ **do**

    Receive stochastic gradients $\mathbf{g}_{\mathcal{W}}^{(t)}, \mathbf{g}_{\mathcal{V}}^{(t)}$ evaluated at $\mathcal{W}^{(t)}$ and $\mathcal{V}^{(t)}$.

    Update accumulators for $\mathcal{Y} \in \{\mathcal{W}, \mathcal{V}\}, \ell \in [2]$: $\mathbf{M}_{\mathcal{Y},\ell}^{(t+1)} = \beta_\ell \mathbf{M}_{\mathcal{Y},\ell}^{(t)} + \mathbf{g}_{\mathcal{Y}}^{(t)}$.

    Compute gradient oracles for $Y \in \{V, W\}$: $\mathbf{A}_{\mathcal{Y}}^{(t+1)} = \mathbf{M}_{\mathcal{Y},1}^{(t+1)} / \sqrt{\mathbf{M}_{\mathcal{Y},2}^{(t+1)} + \varepsilon}$.

    Update: $\mathcal{W}^{(t+1)} = \mathcal{W}^{(t)} + \eta_D^{(t)} \mathbf{A}_{\mathcal{W}}^{(t+1)}, \qquad \mathcal{V}^{(t+1)} = \mathcal{V}^{(t)} - \eta_G^{(t)} \mathbf{A}_{\mathcal{V}}^{(t+1)}$.

**return** $\mathcal{W}^{(T)}, \mathcal{V}^{(T)}$.

---

networks: the generator maps points in the latent space $\mathcal{D}_z$ to sample candidates of the desired distribution; the discriminator evaluates these samples by comparing them to samples from $\mathcal{D}$.

More formally, the generator is a mapping $G_{\mathcal{V}} \colon \mathbb{R}^k \to \mathbb{R}^d$ where $\mathcal{V}$ is some parameter set. Generally, the latent variables are sampled from the normal distribution. On the other hand, the discriminator is a mapping $D_{\mathcal{W}} \colon \mathbb{R}^d \to \mathbb{R}$ where $\mathcal{W}$ is some parameter set. In this section, one can think of these parameter sets as made of matrices and vectors. To train the model, we consider the WGAN-GP problem formulation (Gulrajani et al., 2017) (where $\widehat{\mathcal{D}} = \epsilon \mathcal{D} + (1-\epsilon)\mathcal{D}_z$ for $\epsilon > 0$),

$$\min_{\mathcal{V}} \max_{\mathcal{W}} \ \mathbb{E}_{z \sim \mathcal{D}_z}[D_{\mathcal{W}}(G_{\mathcal{V}}(z))] - \mathbb{E}_{x \sim \mathcal{D}}[D_{\mathcal{W}}(x)] + \lambda \mathbb{E}_{y \sim \widehat{\mathcal{D}}}[(\|\nabla_{\mathcal{W}} D_{\mathcal{W}}\|_2 - 1)^2] := f(\mathcal{V}, \mathcal{W}). \quad (1)$$

**Adaptive methods.** In this paper, we particularly focus on second-moment-based adaptive optimizers to solve (1). Figure 1 captures the usual formulations of Adam (Kingma & Ba, 2014), Adagrad (Duchi et al., 2011) and RMSprop (Tieleman et al., 2012) up to some scaling conventions that can be absorbed into $\{(\eta_G^{(k)}, \eta_D^{(k)})\}$. Note that all operations on vectors in Figure 1 are entry-wise and $\mathbf{g}^2$ denotes the entry-wise power on vector $\mathbf{g}$.

The exponential window parameters $\beta_1, \beta_2 \in [0, 1)$ respectively denote the first- and second-order momentum parameters. A rule of thumb is to set them as $\beta_1 = 0.5$ and $\beta_2 = 0.9$. As we aim to understand the role of adaptivity, an important baseline is stochastic gradient descent ascent with momentum (SGDA) which consists in the update of Figure 1 when the denominator $\sqrt{M_{\mathcal{Y},2} + \epsilon}$ is omitted. *In what follows, we refer to SGDA as the SGDA with momentum update.*

Optimizers as autorefalg:adaptive are *adaptive* in that they keep updating step-sizes while training the model. Therefore, two types of schedules are involved: the *explicit step size schedules* $\{(\eta_G^{(t)}, \eta_D^{(t)})\}$ that the practitioner manually sets up and the *implicit step size schedules* induced by the optimizers. The contributions of these two schedules overlap and it remains unclear how they contribute to the superior performance of adaptive methods relative to SGDA in GANs.

**Method dissection using grafting.** To decouple the explicit and implicit schedules, we adopt the step size grafting approach proposed by Agarwal et al. (2020) and described as follows. At each iteration, we compute stochastic gradients, pass them to two optimizers $\mathcal{A}_1, \mathcal{A}_2$ and make a grafted step which combines the *magnitude* of $\mathcal{A}_1$'s step and *direction* of $\mathcal{A}_2$'s step. Since our goal is to understand the benefits of adaptivity, it is natural to consider the grafting approach with Adam and SGDA. We define *AdaLR*, an optimizer that updates in the SGDA direction with Adam magnitude:

$$\mathcal{W}^{(t+1)} = \mathcal{W}^{(t)} + \eta_D^{(t)} \|\mathbf{A}_{\mathcal{W}}^{(t)}\|_2 \frac{\mathbf{M}_{\mathcal{W},1}^{(t)}}{\|\mathbf{M}_{\mathcal{W},1}^{(t)}\|_2 + \varepsilon}, \ \mathcal{V}^{(t+1)} = \mathcal{V}^{(t)} - \eta_G^{(t)} \|\mathbf{A}_{\mathcal{V}}^{(t)}\|_2 \frac{\mathbf{M}_{\mathcal{V},1}^{(t)}}{\|\mathbf{M}_{\mathcal{V},1}^{(t)}\|_2 + \varepsilon}, \quad (2)$$

and *AdaDir* which updates in the Adam direction with SGDA magnitude

$$\mathcal{W}^{(t+1)} = \mathcal{W}^{(t)} + \eta_D^{(t)} \|\mathbf{M}_{\mathcal{W},1}^{(t)}\|_2 \frac{\mathbf{A}_{\mathcal{W}}^{(t)}}{\|\mathbf{A}_{\mathcal{W}}^{(t)}\|_2 + \varepsilon}, \ \mathcal{V}^{(t+1)} = \mathcal{V}^{(t)} - \eta_G^{(t)} \|\mathbf{M}_{\mathcal{V},1}^{(t)}\|_2 \frac{\mathbf{A}_{\mathcal{V}}^{(t)}}{\|\mathbf{A}_{\mathcal{V}}^{(t)}\|_2 + \varepsilon}. \quad (3)$$

Note that two implementations are possible for AdaLR and AdaDir. In the *layer-wise* version, $\mathcal{Y}^{(t)}$ is a single parameter group (typically a layer in a neural network) and therefore, the update is applied to each group. In the *global* version, $\mathcal{Y}^{(t)}$ contains all of the model's weights. Note that in Figure 2, we set up AdaLR and AdaDir with the layer-wise version –the global AdaLR and AdaDir perform approximately as well as their layer-wise counterparts.

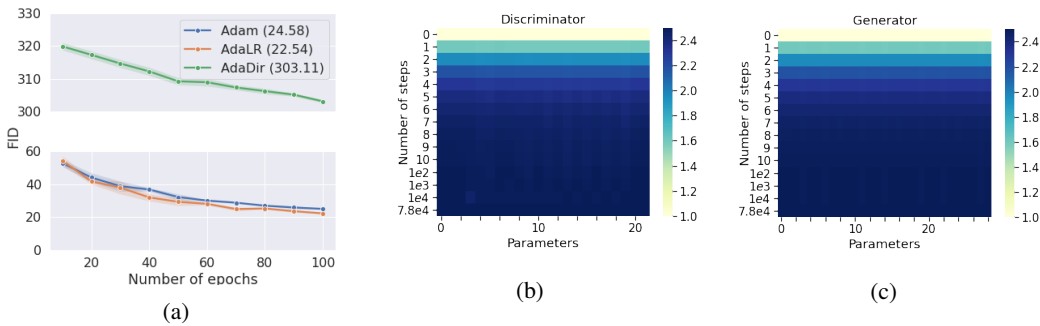

(a)      (b)      (c)

Figure 2: (a) FID score against the number of epochs of a Resnet WGAN-GP trained on CIFAR-10 with Adam, AdaLR, and AdaDir. AdaLR performs slightly better than Adam while AdaDir performs very poorly. (b)-(c) displays the fluctuations of AdaLR's adaptive magnitude. We plot the ratio $\|\mathbf{A}_{\mathcal{Y}}^{(t)}\|_2/\|\mathbf{A}_{\mathcal{Y}}^{(0)}\|_2$ for each discriminator's (b) and generator's (c) parameters. At early stages, this ratio barely increases and remains constant after 10 steps. We train for 100 epochs using a batch-size 64 and results are averaged over 5 seeds.

We train a WGAN-GP (Arjovsky et al., 2017) on CIFAR-10 with AdaLR, AdaDir and Adam. Figure 2(a) shows the GAN performance measured in FID score (Heusel et al., 2017) obtained by the three trained models. We observe that AdaDir does not generate samples from the desired distribution as its FID score is high. On the other hand, AdaLR performs slightly better than Adam. Therefore, we deduce that the success of adaptive methods in GANs may be explained by the *implicit step-size schedule* induced by the algorithm.

**From AdaLR to normalized SGDA.** The previous experiment hints that AdaLR which combines Adam's magnitude and SGDA direction performs better than Adam. We here take a closer look at this algorithm. In particular, we investigate the fluctuations of Adam's magnitude during training. Figures 2(b),2(c) show that this magnitude barely varies for both generator and discriminator.

Therefore, an update with SGDA direction and constant step-size seems to be a valid approximation of Adam in GAN training. Such an algorithm corresponds to normalized SGDA (nSGDA),

$$\mathcal{W}^{(t+1)} = \mathcal{W}^{(t)} + \eta_D^{(t)} \frac{\mathbf{M}_{\mathcal{W},1}^{(t)}}{\|\mathbf{M}_{\mathcal{W},1}^{(t)}\|_2+\varepsilon}, \qquad \mathcal{V}^{(t+1)} = \mathcal{V}^{(t)} - \eta_G^{(t)} \frac{\mathbf{M}_{\mathcal{V},1}^{(t)}}{\|\mathbf{M}_{\mathcal{V},1}^{(t)}\|_2+\varepsilon}. \qquad (4)$$

Similarly to AdaLR and AdaDir, we also define two versions of nSGDA: layer-wise nSGDA (l-nSGDA) and global nSGDA (g-nSGDA). We use both of these in the numerical experiments. As a comparison to (4), the SGDA (with momentum) update is:

$$\mathcal{W}^{(t+1)} = \mathcal{W}^{(t)} + \eta_D^{(t)}\mathbf{M}_{\mathcal{W},1}^{(t)}, \qquad \mathcal{V}^{(t+1)} = \mathcal{V}^{(t)} - \eta_G^{(t)}\mathbf{M}_{\mathcal{V},1}^{(t)}. \qquad (5)$$

## 3 NUMERICAL PERFORMANCE OF NSGDA

Section 2 indicates that nSGDA may work as well as Adam in GAN training. In this section, we empirically verify this hypothesis through an extensive numerical study. To evaluate the proposed algorithm, we conducted extensive experiments on CIFAR-10 (Krizhevsky et al., 2009), LSUN Churches (Yu et al., 2016), STL-10 (Coates et al., 2011) and Celeba-HQ (Liu et al., 2015). Similarly to above, we choose the Fréchet Inception distance (FID) (Heusel et al., 2017) to quantitatively assess the performance of the model. In all our experiments, 50k samples are randomly generated for each model to compute the FID. As for the architectures, we choose Resnets (He et al., 2016) from Gidel et al. (2018) and set up the WGAN-GP loss (Gulrajani et al., 2017). Note that for each optimizer, we grid-search over stepsizes to find the best one in terms of FID. Due to limited computational resources, we trained the models for 100 epochs in the case of CIFAR-10 and Celeba-HQ

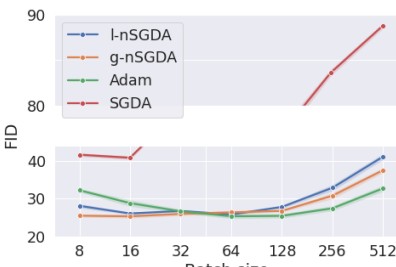

Figure 3: FID score of a Resnet WGAN-GP model trained with l-nSGDA, g-nSGDA, Adam, and SGDA against the batch size in the CIFAR-10 dataset. At small batch sizes, the best performing models are those trained with nSGDA methods. As the batch size increases, the performance of nSGDA methods worsens and Adam performs better. Lastly, models trained with nSGDA consistently outperform those trained with SGDA.

and for 50 epochs for LSUN and STL-10. We apply a linear decay learning rate scheduling during training. All the results are averaged over 10 seeds.

**nSGDA competes with Adam.** Figure 4 shows the performance of l-nSGDA, g-nSGDA, Adam, and SGD on different datasets. We first observe that l-nSGDA and g-nSGDA compete with Adam

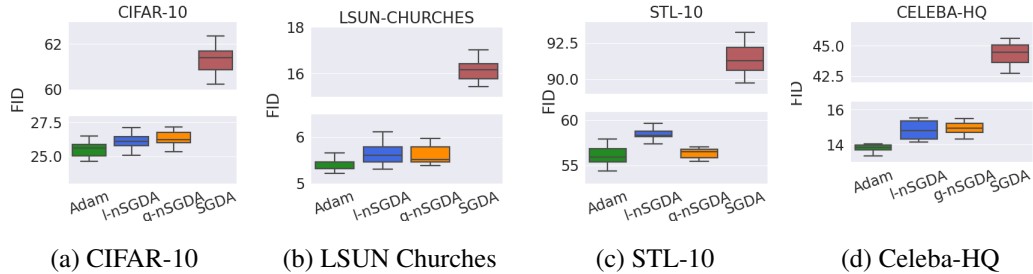

(a) CIFAR-10      (b) LSUN Churches      (c) STL-10      (d) Celeba-HQ

Figure 4: FID scores obtained when training a Resnet WGAN-GP using Adam, l-nSGDA, g-nSGDA, and SGD on different datasets. In all these datasets, l-nSGDA, g-nSGDA and Adam perform approximately as well. SGDA performs much worse. The models are trained with batch-size 64.

in all these datasets. On the other hand, SGD struggles to produce good quality images compared to the other methods as expected. It is worth reminding that the nSGDA methods and SGD have the *same direction* but *differ in their magnitude*. Therefore, this experiment confirms our hypothesis that adaptive methods outperform SGDA thanks to their implicit step-size schedule. Lastly, we highlight that l-nSGDA and g-nSGDA perform almost as well. This suggests that a global step-size adaptivity is enough to perform well in GANs. Similar observations hold for DCGAN (see Appendix A).

**Influence of batch size.** While the nSGDA methods compete with Adam in GAN, it remains unclear whether such performance is proper to the algorithm or tied to the stochastic noise of the gradients. Figure 3 shows the performance of the model trained with l-nSGDA, g-nSGDA, Adam, and SGD and using a wide range of batch sizes. We remark that the nSGDA methods perform better than Adam when the batch size is small. However, as the batch size increases, their performance seriously deteriorates. On the other hand, the performance of Adam seems to be less sensitive to the change of batch size. As expected, SGD overall performs poorly. nSGDA performs well in a relevant batch-size regime, as many GAN architectures such as DCGAN, WGAN-GP, SNGAN (Miyato et al., 2018), SAGAN (Zhang et al., 2019a) requires small batch sizes for optimal performances.

## 4    WHY DOES NSGDA PERFORM BETTER THAN SGDA IN GANS?

In Section 2, we numerically observed that Adam outperforms SGDA mainly thanks to its adaptive magnitude (implicit step-size schedule). In Section 3, we observed that nSGDA methods – which differs from SGDA only by adaptive magnitude – compete with Adam and significantly surpass SGDA. To the best of our knowledge, there is no *theoretical* result that demonstrates the importance of adaptive magnitude in GANs performance. Thus, we set the following question:

> *From a theoretical perspective, for what distribution learning problem using GANs*
> *does nSGDA perform better comparing to SGDA?*

We devise a simple data generation problem where the target distribution consists of two modes, described as vectors $u_1, u_2$, that are slightly correlated (See Assumption 1). We show that using standard GANs' training objective, with high probability, SGDA – with *any reasonable* [1] *step-size configuration* – only learn distributions that suffer from *mode collapse* i.e. $u_1, u_2$ always show up in the learned distribution simultaneously. Conversely, nSGDA learns the two modes *separately*.

**Notations.** We set the GAN 1-sample loss $L_{\mathcal{V}, \mathcal{W}}^{(t)}(X, z) = \log(D_{\mathcal{W}}^{(t)}(X)) + \log(1 - D_{\mathcal{W}}^{(t)}(G_{\mathcal{V}}^{(t)}(z)))$. $\mathbf{g}_{\mathcal{Y}}^{(t)} = \nabla_{\mathcal{Y}} L_{\mathcal{V}, \mathcal{W}}^{(t)}(X, z)$ is the 1-sample stochastic gradient (without momentum).

### 4.1   SETTINGS

In this section, we consider the classical GAN problem formulation.

$$\min_{\mathcal{V}} \max_{\mathcal{W}} \quad \mathbb{E}_{X \sim \mathcal{D}}[\log(D_{\mathcal{W}}(X))] + \mathbb{E}_{z \sim \mathcal{D}_z}[\log(1 - D_{\mathcal{W}}(G_{\mathcal{V}}(z)))]. \quad (6)$$

In the following paragraphs, we describe each element of our formulation.

**Data distribution $\mathcal{D}$.** The data generation problem consists in having a training dataset with points sampled from a target distribution $\mathcal{D}$. Using this dataset, our goal is to train a model that generates samples with the same statistics as $\mathcal{D}$. The latter is defined as follows.

---

[1] Here reasonable simply means that the learning rates are bounded to prevent the training to blow up.

Let $u_1$ and $u_2$ two vectors in $\mathbb{R}^d$ such that $\|u_i\|_2 = 1$, for $i \in \{1, 2\}$.
Each sample from $\mathcal{D}$ consists of an input data $X$ generated as:

1. Sample $s = (s_1, s_2)$ from the distribution $\mathcal{S}$ defined as $s_i \in [0, 1]$, $\mathbb{P}[s_i = 1] \geq 1/2$ for $i \in \{1, 2\}$, such that $\|s\|_0 \geq 1$. Note that $s_1$ and $s_2$ are not necessarily independent.

2. Define data-point $X = s_1 u_1 + s_2 u_2 \in \mathbb{R}^d$.

The distribution $\mathcal{D}$ generates points that are a linear combination of two modes $u_1$ and $u_2$. In what follows, we make the following assumptions on the coefficients $(s_i)$ and modes $(u_i)$.

**Assumption 1** (Data structure). *Let $\gamma = \frac{1}{\text{polylog}(d)}$. The coefficients $s_1, s_2$ and modes $u_1, u_2$ of the distribution $\mathcal{D}$ respect* one *of the following conditions:*

*1. Correlated Modes: $\langle u_1, u_2 \rangle = \gamma$ and the generated data point is either $X = u_1$ or $X = u_2$.*

*2. Correlated Coefficients: $\mathbb{P}[s_1 = s_2 = 1] = \gamma$ and the modes are orthogonal, i.e., $\langle u_1, u_2 \rangle = 0$.*

Assumption 1 captures some of the realistic structure of images. Case 1 models the setting where we have two pure modes (e.g., two different types of cats) that are correlated. Case 2 corresponds to two (roughly) orthogonal modes (e.g., vertical and horizontal edges), that may sometimes be mixed together (images containing object corners in the example above).

**Learner models.** To learn the true distribution $\mathcal{D}$, we use a linear generator $G_\mathcal{V}$ defined as

$$G_\mathcal{V}(z) = Vz = \sum_{i=1}^{m_G} v_i z_i, \tag{7}$$

where $V = [v_1^\top, v_2^\top, \cdots, v_{m_G}^\top] \in \mathbb{R}^{m_G \times d}$ is the weight matrix and $z \in \{0, 1\}^{m_G}$. We set $\mathcal{V} = \{V\}$. Intuitively, $G_\mathcal{V}$ outputs linear combinations of modes $(v_i)$. We assume that $G_\mathcal{V}$ samples from the latent distribution $\mathcal{D}_z$ defined for $z \in \{0, 1\}^{m_G}$, $\|z\|_0 \geq 1$ as:

$$\forall i, j \in [m_G], \ \Pr[z_i = 1] = \Theta\left(\frac{1}{m_G}\right), \quad \Pr[z_i = z_j = 1] = \frac{1}{m_G^2 \text{polylog}(d)} \tag{8}$$

First, remark that vectors sampled from $\mathcal{D}_z$ are binary-valued. Although usual latent distributions in GANs are either Gaussian or uniform, the distribution $\mathcal{D}_z$ should be rather seen as modelling the weights' distribution of a hidden layer of a deep generator. Indeed, Allen-Zhu & Li (2021) argue that the distributions of these hidden layers are sparse, non-negative, and non-positively correlated. Besides, in (8), $\Pr[z_i = z_j = 1] = \frac{1}{m_G^2 \text{polylog}(d)}$ ensures that that the output of the generator is only made of one mode with probability $1 - o(1)$ and thus avoid creating weighted averages of the two or more $v_i$'s (which might cause mode collapsing). To assess the distribution learned by $G_\mathcal{V}$, we also train a non-linear neural network as discriminator $D_\mathcal{W}$

$$D_\mathcal{W}(X) = \text{sigmoid}\left(a \sum_{i \in [m_D]} \sigma(\langle w_i, X \rangle) + \lambda b\right), \quad \sigma(z) = \begin{cases} z^3 & \text{if } |z| \leq \Lambda \\ 3\Lambda^2 z - 2\Lambda^3 & \text{if } z > \Lambda \\ 3\Lambda^2 z + 2\Lambda^3 & \text{otherwise} \end{cases}, \tag{9}$$

where $W = [w_1^\top, \ldots, w_{m_D}^\top] \in \mathbb{R}^{m_D \times d}$ and $a, b \in \mathbb{R}$ are the trainable parameters of $D_\mathcal{W}$, $\lambda > 0$ is a fixed scaling factor (specified below) and $\Lambda = d^{0.2}$. For simplicity, we set $\mathcal{W} = \{W, a, b\}$.

Besides, $\sigma(\cdot)$ is the truncated degree-3 activation function—it is thus made Lipschitz, which is only needed in the proof to *deal* with the case where the generator is trained much faster than the discriminator. Note that this latter case is uncommon in practice.

Lastly, we choose a cubic activation as it is the smallest polynomial degree for the discriminator that is sufficient: as pointed out in Li & Dou (2020), with linear or quadratic activations, the generator can fool the discriminator by only matching the first and second moments of $\mathcal{D}$. By doing so, the generator *cannot* recover the modes $u_1, u_2$ but only weighted averages of them even at the global optimal solution.

**Algorithms.** We solve the training problem (6) using SGDA and nSGDA *with momentum = 0*. For simplicity, we also set the batch-size to 1. At each step $t$, we sample $X \sim \mathcal{D}$ and $z \sim \mathcal{D}_z$ and define SGDA's update as

$$\mathcal{W}^{(t+1)} = \mathcal{W}^{(t)} + \eta_D \mathbf{g}_\mathcal{W}^{(t)}, \qquad \mathcal{V}^{(t+1)} = \mathcal{V}^{(t)} - \eta_G \mathbf{g}_\mathcal{V}^{(t)}, \tag{SGDA}$$

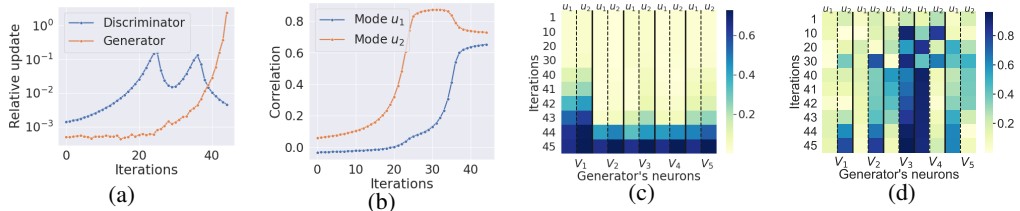

Figure 5: Learning process of SGDA (a,b,c) and learned modes by nSGDA (d) with $m_D = m_G = 5, d = 1000$. We see that in the nSGDA model, each neuron solely learns one of the modes.

where $\eta_D, \eta_G > 0$ are constant step-sizes. On the other hand, nSGDA is defined by the update rule

$$\mathcal{W}^{(t+1)} = \mathcal{W}^{(t)} + \eta_D \frac{\mathbf{g}_{\mathcal{W}}^{(t)}}{\|\mathbf{g}_{\mathcal{W}}^{(t)}\|_2}, \qquad \mathcal{V}^{(t+1)} = \mathcal{V}^{(t)} - \eta_G \frac{\mathbf{g}_{\mathcal{V}}^{(t)}}{\|\mathbf{g}_{\mathcal{V}}^{(t)}\|_2}. \qquad \text{(nSGDA)}$$

Compared to (4), (nSGDA) is a global nSGDA update (without momentum) (i.e. we do not consider *layer-wise* normalization). Indeed, $\|\mathbf{g}_{\mathcal{V}}^{(t)}\|_2$ in the update refers to the sum of norms of the gradients with respect to $a, b, W$. We now detail how to set parameters in (SGDA) and (nSGDA).

**Parametrization 4.1.** *When running SGDA and nSGDA on (1), we set the parameters as*

– **Initialization**: $b^{(0)} = 0$, $a^{(0)} \sim \mathcal{N}\left(0, \frac{1}{m_D \operatorname{polylog}(d)}\right)$, $w_i^{(0)} \sim \mathcal{N}\left(0, \frac{1}{d}\mathbf{I}\right)$, $v_j^{(0)} \sim \mathcal{N}\left(0, \frac{1}{d^2}\mathbf{I}\right)$ *for $i \in [m_D]$, $j \in [m_G]$.*

– **Number of iterations**: *we run SGDA for $t \leq T_0$ iterations where $T_0$ is the first iteration such that $\|\nabla \mathbb{E}[L_{\mathcal{V}^{(T_0)}, \mathcal{W}^{(T_0)}}(X, z)]\|_2 \leq 1/\operatorname{poly}(d)$. . For nSGDA, we run for $T_1 = \tilde{\Theta}\left(\frac{1}{\eta_D}\right)$ iterations.*

– **Step-sizes**: *For SGDA, $\eta_D, \eta_G \in (0, \frac{1}{\operatorname{poly}(d)})$. For nSGDA, $\eta_D \in (0, \frac{1}{\operatorname{poly}(d)}]$, $\eta_G = \frac{\eta_D}{\operatorname{polylog}(d)}$.*

– **Over-parametrization**: *For SGDA, $m_D, m_G = \operatorname{polylog}(d)$ are arbitrarily chosen i.e. $m_D$ may be larger than $m_G$ or the opposite. For nSGDA, we set $m_D = \log(d)$ and $m_G = \log(d) \log \log d$.*

Parametrization 4.1 corresponds to usual initialization and optimization hyper-parameters. Regarding initialization, the discriminator's weights are sampled from a standard normal and its bias is set to zero. The weights of the generator are initialized from a normal with variance $1/d^2$ (instead of the $1/d$ in standard normal). Such a choice is explained as follows. In practice, the target $X \sim \mathcal{D}$ is a 1D image, thus has entries in $[0, 1]^d$ and norm $O(\sqrt{d})$. Yet, we sample the initial generator's weights from $\mathcal{N}(0, \mathbf{I}_d/d)$ in this case. In our case, since $\|u_i\|_2 = 1$, the target $X = s_1 u_1 + s_2 u_2$ has norm $O(1)$. Therefore, we scale down the variance in the normal distribution by a factor of $1/d$ to match the configuration encountered in practice. Therefore, we also set $\lambda = \frac{1}{\sqrt{d}\operatorname{polylog}(d)}$ in (9) to ensure that the weights and the bias in the discriminator learn at the same speed.

Regarding the number of iterations, our theorem holds when running SGDA for any (polynomially) possible number of iterations –after $T_0$ steps, the gradient becomes inverse polynomially small and SGDA essentially stops updating the parameters. Lastly, we allow any step-size configuration for SGDA i.e. larger, smaller, or equal step-size for $D$ compared to $G$. To our knowledge, this setting has never been studied in GAN performance. Note that our choice of step-sizes for nSGDA matches with the one used in practice i.e. $\eta_D$ slightly larger than $\eta_G$.

### 4.2 MAIN RESULTS

We state our main results on the performance of models trained using (SGDA) and (nSGDA). We show that nSGDA is able to learn the two modes of the distribution $\mathcal{D}$ while SGDA is not.

**Theorem 4.1** (SGDA suffers from mode collapse). *Let $T_0, \eta_G, \eta_D$ and the initialization as defined in Parametrization 4.1. Let $t$ be such that $t \leq T_0$. Run SGDA for $t$ iterations with step-sizes $\eta_G, \eta_D$. Then, with probability at least $1 - o(1)$, for all $z \in \{0, 1\}^{m_G}$, we have:*

$$G_{\mathcal{V}}^{(t)}(z) = \alpha^{(t)}(z)(u_1 + u_2) + \xi^{(t)}(z), \quad \text{where } \alpha^{(t)}(z) \in \mathbb{R} \text{ and } \xi^{(t)}(z) \in \mathbb{R}^d,$$

*such that for all $\ell \in [2]$, $|\langle \xi^{(t)}(z), u_\ell \rangle| = o(1)\|\xi^{(t)}(z)\|_2$ for every $z \in \{0, 1\}^{m_G}$.*

*In the specific case where $\eta_G = \frac{\sqrt{d}\eta_D}{\operatorname{polylog}(d)}$, the model mode collapses i.e. $\|\xi^{(T_0)}(z)\|_2 = o(\alpha^{(T_0)}(z))$.*

Theorem 4.1 indicates that when using SGDA with any step-size configuration, the generator either does not learn the modes at all – when $\alpha^{(t)}(z) = 0$, $G_{\mathcal{V}}^{(t)}(z) = \xi^{(t)}(z)$ – or learns an average of the

modes – when $\alpha^{(t)}(z) \neq 0$, $G_{\mathcal{V}}^{(t)}(z) \approx \alpha^{(t)}(z)(u_1 + u_2)$. We emphasize that the theorem holds *for any* time $t \leq T_0$ which is the iteration where SGDA converges to an approximate first-order locally optimal min-max equilibrium. Conversely, nSGDA succeeds to learn the two modes separately.

**Theorem 4.2** (nSGDA recovers modes separately). *Let $T_1$, $\eta_G, \eta_D$ and the initialization as defined in Parametrization 4.1. Run nSGDA for $T_1$ iterations with step-sizes $\eta_G, \eta_D$. Then, the generator learns both modes $u_1, u_2$ i.e.,*

$$\Pr_{z \sim \mathcal{D}_z} \left( \left\| \frac{G_{\mathcal{V}}^{(T_1)}(z)}{\|G_{\mathcal{V}}^{(T_1)}(z)\|_2} - u_\ell \right\|_2 = o(1) \right) = \tilde{\Omega}(1), \quad for \quad \ell = 1, 2. \tag{10}$$

Theorem 4.2 indicates that when we train a GAN with nSGDA in the regime where the discriminator updates slightly faster than the generator (as done in practice), the generator successfully learns the distribution containing the direction of both modes.

### 4.3 WHY DOES SGDA SUFFER FROM MODE COLLAPSE?

We now sketch the reason why SGDA suffers from mode collapse using Figure 5. Figure 5(a) displays the relative update speed $\frac{\eta\|\mathbf{g}_{\mathcal{V}}^{(t)}\|_2}{\|\mathcal{Y}^{(t)}\|_2}$, Figure 5(b) the correlation $\frac{\langle w_i^{(t)}, u_\ell \rangle}{\|w_i^{(t)}\|_2}$ between $D$'s neuron and mode $u_\ell$ and Figure 5(c) the correlation $\frac{\langle v_j^{(t)}, u_\ell \rangle}{\|v_j^{(t)}\|_2}$ between $G$'s neuron and mode $u_\ell$.

We focus on the case where the discriminator's step-size is set so it updates slightly faster than the generator at the beginning – as displayed in iterations 1–5 in Figure 5(a). The key observation is that in the early stages, the discriminator's update is approximately

$$w_i^{(t+1)} \approx w_i^{(t)} + \eta_D \langle w_i^{(t)}, X \rangle^2 X, \qquad \text{for } i \in [m_D]. \tag{11}$$

With high probability, there exists at least a neuron $i$ such that $\langle w_i^{(t)}, X \rangle > 0$. Thus, (11) implies that $\langle w_i^{(t)}, X \rangle$ is an increasing sequence. As $t$ increases, $w_i^{(t)}$ gradually grows its correlation with one of the modes $u_\ell$ (iterations 1-20 in Figure 5(b)) and $D$'s gradient norm thus increases. Therefore, after a first phase where $D$ updates slightly faster than $G$, $D$'s update speed becomes significantly larger than $G$'s one (iterations 5–20 in Figure 5(a)). Thus, $D$ learns the first mode after 20 iterations in Figure 5(b).

However, one of the goal of $D$ (from the training objective in (6)) is to maximize its *average* correlation with the target distribution $\mathcal{D}$. Since $G$ does not catch up and $\mathcal{D}$ is made of two modes showing up with equal probability, the optimal solution for $D$ is to have each of its $w_i = \alpha_i(u_1 + u_2)$ to maximize the average correlation. Therefore, after iteration 20, $D$ learns the second mode and eventually gets this optimal solution (which is a mode collapse) at iteration 40. Then, $D$'s update speed starts dropping as we see in Figure 5(a) which helps $G$ to catch up and grow its update speed. However, since $D$ already learnt a weighted average of the modes, it can only teach $G$ to learn this average and thus mode collapses as we see in Figure 5(c).

On the other hand, nSGDA ensures that $G$ and $D$ *always learn at the same speed*, so that $G$ can learn one mode immediately when $D$ learns (such as at iteration 25 in (b) in Figure (5)), which avoids mode collapse.

## 5 CONCLUSION

Our work offers a complementary view to several works in the min-max optimization literature where the discriminator is much faster than the generator to converge to an equilibrium. Here, instead, we advocate the use of balanced updates to ensure that the GAN performs well.

Our work is a first step towards understanding how adaptive methods improve the GAN performance. Our empirical observations and theorems heavily rely on the fact that the batch size is small. However, nSGDA methods seem to not work well for large batch sizes and may not be suitable to some large-scale GANs such as BigGAN (Brock et al., 2018). It would be interesting to understand why adaptive methods are crucial in this case. Another interesting direction would be to improve the training of nSGDA in the batch setting.

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
