# OpenReview forum: "Adam is no better than normalized SGD:  Dissecting how adaptivity improves GAN performance"
_ICLR.cc/2022/Conference — ICLR 2022 Submitted_

### Official Review · Reviewer_QjHh · 2021-10-28

**Correctness:** 4
**Technical Novelty And Significance:** 4
**Empirical Novelty And Significance:** 4
**Recommendation:** 8
**Confidence:** 4

**Main Review:**

This paper presents a great study that dissects the success of Adam in GAN training, which covers insightful observations and comprehensive analysis both empirically and theoretically. The findings of this paper, such as "it is the adaptive magnitude of Adam that matters" and "the learning objective of GAN converges does not imply the generator synthesizes high-quality samples" will definitely encourage more exploration along this direction, helping us better understand the optimization process of similar non-convex non-concave min-max problems.

Minor concerns:
1. The space is overly adjusted. Related work should be included in a separate section.
2. I suggest avoiding representing both distribution and distriminator as D, which hurts the readability significantly.
3. typos, e.g. autorefalg:adaptive?
4. While the finding of nSGDA is indeed interesting and insightful, it performs comparably with Adam. Can authors discuss the potential advantages of nSGDA over Adam?


**Summary Of The Paper:**

This paper dissects the success of Adam in GAN training by swap the gradient direction/magnitude with that of SGDA. By doing so, authors found the Adam produces higher quality solutions relative to SGDA in GANs mainly due to its adaptive magnitude and not to its adaptive direction. Inspired by this observation, the combination of direction of SGDA and magnitude of Adam yields the use of normalized SGDA (nSGDA) in GAN training, which consistently competes with Adam.

**Summary Of The Review:**

I enjoy reading this paper. The findings are insightful with comprehensive empirical and theoretical analysis.

---

> ### Author Response · Authors · 2021-11-12
> **Response to Reviewer QjHh**
>
>
> We thank Reviewer QjHh for his/her review. We appreciate their positive feedback and hope that they will keep supporting our work along the discussion.
>
> Also, thanks for noticing these typos, we will correct them in the final version.
>
> -- **Advantages of nSDGA over Adam.**
>
> We develop here the advantages of nSGDA over Adam:
>    * **Less tuning:** Adam has 6 parameters to tune (learning rate first and second moment for each player) while our proposed method has only two parameters to tune (2 learning rates).
>    * **Less memory requirement:** nSGDA presents computational advantages compared to Adam. It does not require to maintain a second order momentum buffer as we only need to divide by the norm of the gradient oracle. Thus allowing for faster training.
>    * **Easier Understanding of the method:** The method is very close to SGD which is a method that is well understood and easier to analyze than Adam. Thus, using nSGDA is a step forward for a better understanding of game optimization.
>    * **Theoretical guarantees:**  On the one hand, Adam provably does not converge [1] in the minimization case. Thus, there is no hope to analyze it in the context of differentiable games either. On the other hand, nSGD is a relatively well understood method in minimization [2] and there is high hope to extend its analysis to nSGDA.
>   * **Easier follow-up:** Also it is easier to build on top of a simple method. For instance, analyzing and understanding the impact of extragradient and optimistic methods on nSGDA is significantly simpler than analyzing their impact on Adam. To illustrate this point many methods built on top of Adam such as ExtraAdam [3] and Optimistic-Adam [4] are not analyzed.
>
> Overall our points could be summarized by the sentence **simpler with similar performances is better**. We also would like to point out one more time that with a simple batch-size tuning we improve the performance of our model over Adam (see Figure 4).
>
> [1] Reddi, Sashank J., Satyen Kale, and Sanjiv Kumar. "On the Convergence of Adam and Beyond." International Conference on Learning Representations. 2018.
>
> [2] Hazan, Elad, Kfir Y. Levy, and Shai Shalev-Shwartz. "Beyond convexity: Stochastic quasi-convex optimization." arXiv preprint arXiv:1507.02030 (2015).
>
> [3] Gidel, Gauthier, et al. "A Variational Inequality Perspective on Generative Adversarial Networks." International Conference on Learning Representations. 2019.
>
> [4] Daskalakis, Constantinos, et al. "Training GANs with Optimism." International Conference on Learning Representations (ICLR 2018). 2018.

---

### Official Review · Reviewer_5J1v · 2021-11-02

**Correctness:** 3
**Technical Novelty And Significance:** 2
**Empirical Novelty And Significance:** 2
**Recommendation:** 5
**Confidence:** 3

**Main Review:**

Carefully adjusted update steps in Reinforcement Learning (actor/critic models) and GAN training are vital to reach convergence. E.g. [1] proposed a control variable to level out discriminator and generator. [2] has proven that under some assumptions competing agents converge to an equilibrium if the learning speeds are carefully adjusted. [3] based on the results of [2] have shown that the results are also valid in the GAN context where the slower agent is the generator and the faster agent the discriminator. To reach convergence a faster leader, critic, discriminator guides a slower follower, generator such that the follower, generator is fast enough to follow but in the same time is not too fast to disturb the leader [4]. My main concern with this paper is, that having normalized gradient updates is probably a good starting point to select optimal update steps but according to the research mentioned above it's more likely normalized gradient updates still need to be scaled (for both agents separately).
As the learning speeds of the agents depend not only on the learning rates, e.g. the complexity of the architecture is important as well, one has to be careful to select the correct settings. E.g. in the DCGAN architecture the generator has more parameters and needs a higher learning rate compared to the discriminator to converge cf. Section A5 in [3]. However, in the first experiment the authors train the DCGAN generator with a smaller learning rate which probably leads not to optimal FIDs. I suggest to run additional experiments with higher learning rates for the generator and add them to the results shown in Figure 1. In section 4 i'm not sure if i can see a theoretical proof of theorems 4.1 and 4.2.

[1] Berthelot et al., BEGAN: Boundary Equilibrium Generative Adversarial Networks,  https://arxiv.org/abs/1703.10717 balance
[2] V. S. Borkar. Stochastic approximation with two time scales. Systems & Control Letters,
29(5):291–294, 1997
[3] Heusel et al. GANs trained by a two time-scale update rule converge to a local nash equilibrium. Advances in
neural information processing systems, 30, 2017.
[4] M. W. Hirsch. Convergent activation dynamics in continuous time networks. Neural Networks,
2(5):331–349, 1989.
[5] Tanner Fiez and Lillian Ratliff. Gradient descent-ascent provably converges to strict local minmax
equilibria with a finite timescale separation. arXiv preprint arXiv:2009.14820, 2020.

**Summary Of The Paper:**

In this manuscript the authors investigate the effect of normalized and unormalized gradient updates on the convergence of GANs. In a first experiment a DCGAN model is trained with Adam and it is shown that the best models evaluated with the FID have balanced learning rates but haven't converged while models trained with extremly unbalanced learning rates converge while having high FIDs i.e. didn't learn the training distribution.
In the second experiment it is shown that an adaptive gradient magnitude helps to train a good model compared to an adaptive gradient direction. In a next step using the WGAN-GP model on CIFAR-10, LSUN Churches, STL-10, and CelebA-HQ datasets the optimizers Adam, normalized gradient updates and unnormalized SGDA were compared. The normalized optimizers including Adam outperformed SGDA. Finally it was shown that SGDA is more sensitive to the batch-size. In a final experiment on a small toy dataset it was shown that SGDA suffers from mode collapse while normalized gradient updates guide the generator to learn the modes of the dataset.


**Summary Of The Review:**

The importance of balanced updates of the generator and discriminator in GAN training is already well known. Using normalized gradient update steps for convergence is plausible, however not enough as shown in [2,3,4,5].

---

> ### Author Response · Authors · 2021-11-12
> **Response to Reviewer 5J1v**
>
> We thank the reviewer 5J1v for his/her review. We would like to clarify some points in the paper to ensure that the purpose of the paper is well-understood.
>
> -- **Clarification of the goal of the paper**.
>
> We recall that the criterion we track is **not convergence but rather performance** (measured in FID score in the experiments).
> We numerically observe over several real world datasets that scaling the gradient updates (as in normalized SGDA) improves model performance, when compared with a model trained with vanilla SGDA (in terms of image quality). We prove this observation in a simplified setting.
>
> -- **Comparison of our work with [1], [2].**
>
> [1] shows that Adam converge to a Nash equlibrium. However, this work is very different from ours: first, they look at the convergence of the algorithm while we are interested in the performance. In Figure 1a), DCGAN does not converge and yet, find a good-quality solution (small FID score). We believe that convergence may not be able to explain performance. Besides, [1] assume that $a_n=o(b_n)$, where $a_n$ is G's and $b_n$  is D's stepsize to show convergence.  These two-time scale stepsizes do not correspond to what is mostly done in practice [3,4].
> In [2], the authors modify the objective loss and observe that the balancedness between D and G yields better performance. Our approach is different as we do not change the original GAN objective. We empirically observe that contrary to SGDA, Adam and nSGDA manage to keep balanced updates.
> We thank the reviewer for these relevant references on the need of carefully adjusted learning rates to converge to an equilibrium. In our updated version of the paper, we included this comparison in the related work section.
>
> -- **"nSGDA is not good enough".**
>
> **We are confused by the critics of the reviewer regarding nSGDA**. We show that nSGDA is good enough for training GANs over many real-world data sets. We also prove that it is better than SGDA in a analysable GAN learning setting. nSGDA with a careful tuning of the step-size of both players is good (more details in the next answer).
>
> -- **"it's more likely normalized gradient updates still need to be scaled (for both agents separately)."**
>
> We would like to clarify the difference between SGDA and normalized-SGDA (see equation 4 for nSGDA and equation 5 for SGDA). In both algorithms, the learning rate $\eta_G$ of the generator and $\eta_D$ of the discriminator can be tuned before the training, but **they remain fixed during the entire training process, which is what is typically done in practice for training GANs**.
>
> We show numerically and theoretically that **SGDA with any such fixed step-size configuration yields relatively poor performance**. In contrast, we numerically and theoretically show that for nSGDA, there exist such fixed step-size configuration for which the model performs well.
>
> -- "**the first experiment the authors train the DCGAN generator with a smaller learning rate which probably leads not to optimal FIDs. "**
>
> **We did not use small learning rate:** We clarify our setting for the experiment in Figure 1. To produce this plot, we selected 20 step-size ratios ranging from 1 to 100. For each of them, we grid-searched over a very large range of step-sizes (from 5e-5 to 1e-2). We selected the best points in terms of FID score when ratio <= 20. Beyond that, the FID score is very high and so, we selected those which were not diverging.
>
> **We remind the goal of Figure 1**: we want to show that the performance of a GAN is not tied to the convergence. When the step-size ratio is extremely large, the algorithm manages to converge to an (first order) local equilibrium on those configurations. On the other hand, for any fixed small step-size ratio, no matter what the learning rates are, the GAN does not converge and yet, the FID score is relatively low. Therefore, convergence of GANs training seems to not be tied to performance.
>
> -- **Ratios smaller than 1 in Figure 1a.**
>
> We would like to emphasize that in standard GAN papers, the learning rate of the discriminator *is always* chosen larger than the one of the generator (see WGAN-GP [3], DCGAN [4] among others) which means that the ratios in Figure 1a reflect what is used in practice. We have run additional experiments and updated Figure 1a with ratios <=1 (see figure in the revision of paper). Note that this reinforces our conclusions.
>
> --  **"i'm not sure if i can see a theoretical proof of theorems 4.1 and 4.2."**
>
> The proof of Theorem 4.1 is in Appendix F.4 and the one of Theorem 4.2 is in Appendix H.5 due to space limit.
>
> [1] Heusel, M., et al. "Gans trained by a two time-scale update rule converge to a local nash equilibrium."
>
> [2] Berthelot, D., et al. "Began: Boundary equilibrium generative adversarial networks."
>
> [3] Gulrajani, I., et al. "Improved training of wasserstein gans."
>
> [4] Radford, A., et al. "Unsupervised representation learning with deep convolutional generative adversarial networks."

---

> > ### Comment · Reviewer_5J1v · 2021-11-21
> > **Response to the authors**
> >
> > I thank the authors for their answers and additional experiments added to Figure 1! However i'm still not fully convinced
> > - As already mentioned [1] have already proven and [2] applied to GANs, that in two-player games with stochastic updates ie. mini-batches as used in practice, the step sizes for the faster leader and the slower follower have to be carefully selected such that the leader converges, the follower can follow but does not disturb the leader too much. The authors argue in the same direction but this insight is then not novel. It's probably the case that for GANs normalized gradients are already a good choice such that the aforementioned conditions are met or met enough. However it's not clear that normalized gradients are always sufficient. As far as I have understood the authors prove only for a specific toy example that mode collapse does not happen with normalized gradients (Theorem 4.2)?
> > - The authors show in Figure 1 that successfully trained models defined by a low FID and visual inspection are not converging and even have very high gradients ie are not in equilibrium. In my understanding this this finding is very remarkable and in my opinion the most interesting part of the paper.
> > - In Figure 1 b i don't understand the two curves for the generator and discriminator, shouldn't it be one curve with the gradient ratio of both? What are the numbers in the round brackets?
> > - In Figure 5 for SGDA the learning process with mode collapse is shown. It would be great to see the same figure/experiments with nSGDA to show that mode collapsing is indeed not happening and D and G play at the same speed as claimed by the authors.
> >
> > [1] V. S. Borkar. Stochastic approximation with two time scales. Systems & Control Letters, 29(5):291–294, 1997
> >
> > [2] Heusel et al. GANs trained by a two time-scale update rule converge to a local Nash equilibrium. Advances in
> > neural information processing systems, 30, 2017.

---

> > > ### Author Response · Authors · 2021-11-21
> > > **Response to Reviewer 5J1v**
> > >
> > >
> > > We thank the reviewer for their response.
> > >
> > > --- **"[2] already proved that the stepsizes of faster leader and the slower follower have to be carefully selected such that the leader converges."**
> > >
> > > As explained in the main paper and in our response to the reviewer, our paper does not consider at all convergence. Once again, we believe that convergence is totally disentangled from performance in GANs. Figure 1a  empirically shows this. We see that all the models that have a low FID score do not have a zero gradient ratio.
> > >
> > > --- **Comparison with [2].**
> > >
> > > As already explained in the response and added in the main paper, we believe that [2] focuses on a completely different scope compared to our work. Here are the reasons:
> > >
> > > -- We do not focus on convergence but rather on performance: the empirical and theoretical results are on performance contrary to [2]. As the reviewer already pointed out, we observed in our paper that convergence might have nothing to do with performance. Thus the convergence results shown [2] might be irrelevant for the GANs performance as we considered in our paper.
> > >
> > > -- In their [2] Assumes that the sum of squares of step-sizes for all iterations are bounded, this is indeed false for normalized SGDA (or even SGDA) with fixed step-sizes -- Which is the popular practical choice of step-sizes when training GANs, and is the setting considered in our work.
> > >
> > > -- [2] develop theoretical results on SGDA (Theorem 1) and Adam on single objective minimization (Theorem 2). It is very unlikely that Adam could be analyzed in the TTUR setting since it has been shown that *Adam does not converge in general* [3]. On the other hand, we analyze **nSGDA** and prove that it is a simpler alternative to Adam for which theoretical advances are more likely to happen.
> > >
> > > -- Theory: To use [1], [2] need the TTUR conditions. As mentioned in the response and in the revised paper, this means that the assumption is that $\eta_G\ll \eta_D$. But Figure 1 shows that this regime cannot empirically produce valid results. Our theoretical result does not make such an assumption: we emphasize the balanced update given by nSGDA where both sides need to LEARN AT SIMILAR SPEED (note that *SGDA and nSGDA updates a fundamentally* different due to the *normalization of each stochastic gradients*). Besides, our result is not a convergence result but rather a generalization result. In fact, we prove that in our setting, SGDA with any learning rates configuration. **indeed converges to a local nash**, but still, fail to learn the two modes of the true distribution.
> > >
> > > --- **Proof for a specific toy example.**
> > >
> > > Our work belongs to the field of algorithmic regularization, where many important papers in this literature make simplified structural assumptions in order to prove theoretical results. Indeed, at the current stage it is unreasonable to expect results like "nSGDA works better than SGDA for all problems in GANS in terms of learning the actual distribution " -- This is way beyond the current level of theory, since learning the target distribution (as in our work) is more relevant to the Global Nash instead of local Nash.  Besides, note that the fact that nSGDA empirically works better than SGDA in GANs also depends on structural assumptions (data, loss,  batch size). Our framework attempts to capture all these assumptions: we use the GAN loss, a small batch size (as used in practice in GANs),  and our data distribution is a simplified setting that captures that real datasets have many modes that are correlated (e.g. two different types of cats).  Lastly, please note that to our knowledge, this is the first work that establishes a separation result between SGDA and another algorithm in terms of generalization in GANs and relies on totally novel proof techniques.
> > >
> > > --- **Clarification of Figure 1b.**
> > >
> > > The discriminator and the generator *have their own parameter* and gradients are defined with respect to the discriminator's parameters and to the generator's parameters. Thus, we plot the gradient ratio for discriminator and generator separately in this plot, to show that neither of them converges (thus the gradient ratio of both cannot either).
> > >
> > > --- **"What are the numbers in the round brackets?"**
> > >
> > > They correspond to the gradient ratio at the last epoch value. This is to enhance that the gradient ratio is still large while the images produced are of good quality.
> > >
> > > --- **"same figure/experiments with nSGDA to show that mode collapsing is indeed not happening and D and G play at the same speed as claimed by the authors."**
> > >
> > > We thank the reviewer for this interesting idea. The goal of the synthetic experiments was initially to explain why SGDA mode collapses. In the revised version, we have just added the experiments when using nSGDA. We indeed see that nSDGA does not mode collapses.
> > >
> > >
> > > [3] Reddi, Sashank J., Satyen Kale, and Sanjiv Kumar. "On the Convergence of Adam and Beyond." International Conference on Learning Representations. 2018.

---

### Official Review · Reviewer_5fFW · 2021-11-03

**Correctness:** 4
**Technical Novelty And Significance:** 4
**Empirical Novelty And Significance:** 3
**Recommendation:** 5
**Confidence:** 4

**Details Of Ethics Concerns:**

None.

**Main Review:**

Strengths:
1. This paper finds that the adaptive magnitude of Adam is critical to the performance of GANs. Personally, I think the analysis of the optimizers for GANs is an interesting topic.
2. In the theoretical aspect, the analysis explains the success of nSGDA.
3. The results show that nSGDA achieves a similar performance compared to Adam using WGAN-GP.

Weaknesses:
1. Although the paper finds that the adaptive magnitude of Adam helps the performance in GANs, the proposed nSGDA is still inferior to Adam. The study does not analyze why Adam is superior to nSGDA. I would expect the study presents how to further improve nSGDA or even presents methods that outperform Adam.
2. The study uses a ResNet WGAN-GP as the network backbone. I would expect to use more recent GANs, such as BigGAN and StyleGAN.
3. For the optimizer, the current experiments are not comprehensive enough. I think the proposed optimizer should be evaluated over different network architectures, different resolutions, and different GANs.


**Summary Of The Paper:**

This paper studies how adaptive methods help performance in GANs. The study empirically finds that SGDA with the same vector norm as Adam reaches similar better performance. Based on this observation, normalized SGDA (nSGDA) is proposed as a simpler alternative to Adam. nSGDA is evaluated on several datasets and the results demonstrate that nSGDA is more stable than SGDA.

**Summary Of The Review:**

Although the topic and findings of this paper are interesting, I think the contributions of the current version are limited, especially for the performance of nSGDA and the experiments.

******************************
Post Rebuttal Comments:

I thank the authors for their efforts in addressing my concerns. However, I still have the major concern about the performance of the proposed method over commonly used GANs such as BigGAN and StyleGAN. The authors respond that simpler with similar performances is better. However, there is no result to show similar performance over BigGAN or StyleGAN. Thus I keep my original rating.

---

> ### Author Response · Authors · 2021-11-12
> **Response to Reviewer 5fFW**
>
>
> We thank Reviewer 5fFW for his/her detailed review. We would like to ensure that the purpose of the paper is well-understood.
>
> -- **nSGDA does not outperform Adam**.
>
> The purpose of this paper is **not** to design an optimizer that outperforms Adam. Instead, we want to understand why adaptive methods outperform **SGDA** in GAN training in order to find **simpler optimizers**. As detailed in Section 2, we empirically show that AdaLR, which updates in the SGDA direction with Adam magnitude outperforms Adam. This suggests that Adam outperforms SGDA mainly thanks to its adaptive magnitude. We then observe that Adam's magnitude stays within a constant range during training which incentivizes to look at the case where the magnitude is constant: this yields nSGDA. In section 3, we see that nSGDA outperforms Adam for small batch sizes and competes with it for larger batch sizes. Thus, our experiments show that the main reason why adaptive methods is the normalization of the gradient.
>
> We develop here the advantages of nSGDA over Adam:
>
>   + **Less tuning**: Adam has 6 parameters to tune (learning rate first and second moment for each player) while our proposed method has only two parameters to tune (2 learning rates).
>   + **Less memory requirement**: nSGDA presents computational advantages compared to Adam. It does not require to maintain a second order momentum buffer as we only need to divide by the norm of the gradient oracle. Thus allowing for faster training.
>    + **Easier Understanding of the method**: The method is very close to SGD which is a method that is well understood and easier to analyze than Adam. Thus, using nSGDA is a step forward for a better understanding of game optimization.
>    + **Theoretical guarantees**:  On the one hand, Adam provably does not converge [4] in the minimization case. Thus, there is no hope to analyze it in the context of differentiable games either. On the other hand, nSGD is a relatively well understood method in minimization [2] and there is high hope to extend its analysis to nSGDA.
>
> Overall our points could be summarized by the sentence **simpler with similar performances is better**. We also would like to point out one more time that with a simple batch-size tuning we improve the performance of our model over Adam (see Figure 4).
>
> -- **Experiments with BigGAN and styleGAN?**
>
> Although those architectures achieve state-of-the-art results in generating images, they rely on several tricks (such as orthogonal regularization, spectral normalization, truncation of the latent etc.) to work well. Those tricks make it hard to isolate the benefit of adaptive algorithms compare to SGDA in terms of training GANs.
>
> --  **Evaluation over different network architectures, different resolutions, and different GANs.**
>
> Our experiments cover a numerous amount of settings: two architectures (WGAN-GP) and (DCGAN) evaluated over 4 datasets (CIFAR-10, LSUN Churches, STL-10, CELEBA-HQ). For each setting, we run the experiment for 4 algorithms (Adam, g-nSGDA, l-nSGDA, SGDA) and the results are averaged over 5 seeds. The results were consistent over these settings and give us insight to derive a theoretically explain why adaptive methods outperform SGDA.
>
> -- **Contributions are limited.**
>
> Our goal is not to show that nSGDA is a new algorithm that works as well as Adam in GANs.
>
> Our goal is to explain why adaptive methods outperform SGDA in GANs. **We are not aware of any existing formal explanation**.
>
> We made the following contributions:
> (1). We observe that the good performance of Adam in GANs is **NOT** associated with whether the training converges or not (See figure 1). In fact, for standard GANs training, Adam can learn good quality solutions (in terms of FID score) without even converging.
> (2). We show that Adam compare to SGDA is better mainly due to its adaptive magnitude, not the update direction. This motivates us to study nSGDA as a simpler version of an optimization algorithm with adaptive magnitude.
>
> (3). Theoretical **This is our main contribution**: We prove in a simplified setting  of GANs that nSGDA learns better quality solutions compare to SGDA, although SGDA can converge to a local first order nash under any learning rate configuration. To the best of our knowledge, most of the theory papers in the GAN literature prove convergence of optimization algorithms while we rather provide a "generalization" result. As we point out in (1) (as well as in many citations in our paper),  the good/bad performance of optimization algorithms in GANs is **NOT** associated with the convergence.  **Our theoretical analysis emphasizes the importance of balanced updates between G and D in GANs, to avoid mode collapsing in distribution learning**, which is the main advantage of adaptive optimization algorithms compare to SGDA.
>
>
>
> [2] Hazan, E., et al. "Beyond convexity: Stochastic quasi-convex optimization." .
>
> [4] Reddi, S., et al. "On the convergence of adam and beyond." .

---

### Decision · Program_Chairs · 2022-01-20

**Decision:**

Reject

**Comment:**

The paper studies how adaptive methods help train GANs to achieve better FID scores. It empirically shows that the adaptive magnitude in ADAM is the reason for ADAM's wide adoption for GAN training. The paper receives three reviews: one ranked the paper "accept, good paper" and two ranked the paper "marginally below the acceptance threshold". The supportive reviewer likes the findings in the paper interesting but does not provide enough explanation on the significance of the findings. On the other hand, the negative reviewers raise several concerns, including the GAN architectures used in the paper are outdated and the achieved performance gain is not major. As the paper focuses on performance instead of convergence, the meta-reviewer feels it would be better to include results on SOTA GAN architectures. The provided rebuttal does not lead to any review score change. Consolidating the review and rebuttal, the meta-reviewer feels the paper needed to be improved to meet the bar and would not recommend its acceptance.